# On Solvability of Some Inverse Problems for a Fractional Parabolic Equation with a Nonlocal Biharmonic Operator

Moldir Muratbekova [1,*,†], Bakhtiyar Kadirkulov [2,†], Maira Koshanova [1,†] and Batirkhan Turmetov [1,†]

1   Department of Mathematics, Khoja Akhmet Yassawi International Kazakh-Turkish University,
    Turkistan 161200, Kazakhstan; maira.koshanova@ayu.edu.kz (M.K.); batirkhan.turmetov@ayu.edu.kz (B.T.)
2   Department of Mathematics and Information Technologies, Tashkent State University of Oriental Studies,
    Tashkent 100060, Uzbekistan; baxtiyar_kadirkulov@tsuos.uz
*   Correspondence: moldir.muratbekova@ayu.edu.kz
†   These authors contributed equally to this work.

**Abstract:** The paper considers the solvability of some inverse problems for fractional differential equations with a nonlocal biharmonic operator, which is introduced with the help of involutive transformations in two space variables. The considered problems are solved using the Fourier method. The properties of eigenfunctions and associated functions of the corresponding spectral problems are studied. Theorems on the existence and uniqueness of solutions to the studied problems are proved.

**Keywords:** inverse problem; fractional derivative; involution; nonlocal operator; parabolic equation; fourth order equation; eigenfunctions and associated functions

## 1. Introduction

In this paper, the solvability of some inverse problems for the fractional analogue of a parabolic equation with a nonlocal biharmonic operator are studied. The nonlocal biharmonic operator is defined with the help of involutive mappings given in the space $R^2$. An involutive mapping or an involution $S$ is a mapping that has the property $S^2 = I$, where $I$ is the identity mapping.

As is well known, studies of problems in which, along with solving the equation, it is necessary to find the right side, the coefficient of the equation, or the initial and boundary functions are called inverse problems of mathematical physics. Inverse problems have numerous applications in modern science; they arise in the study of problems in acoustics, astronomy, geophysics, seismology, medical tomography, and other areas (see, for example, [1,2] and references therein).

Direct and inverse problems for fractional differential equations with involution were studied in [3–10]. In these works, the problems of finding the solution and the right side of the equation in the case of one spatial variable were considered. Inverse problems in the case of two spatial variables for differential equations of fractional order were studied in [11–16].

As far as we know, the initial-boundary value problems for partial differential equations of fractional order with operators of the fourth and higher orders have been insufficiently studied. In this direction, we can note works [17–24] where, in particular, the inverse problems were also studied.

The problems close to our studies were considered in the work of S. Kerbal et al. [25]. In this work, for a differential equation of a fractional order with a differential operator of the fourth order in two spatial variables, an initial-boundary value problem is studied, where the boundary conditions in the first variable are specified as Dirichlet-type conditions, and those in the second variable are considered as nonlocal Samarsky–Ionkin-type conditions. Note that the attention of researchers to the study of the problems of the Samarskii–Ionkin type was initiated by the publication of N.I. Ionkin's papers [26,27]. In contrast to classical problems, in this problem, the corresponding spatial differential operator is non-self-adjoint,

and, therefore, the system of eigenfunctions is incomplete. Consequently, problems arise in studying the completeness and basis property of such systems. In [25], this problem was studied for the fourth-order differential operators.

In this paper, we study two types of problems for a fractional-order differential equation with a nonlocal biharmonic operator. In the first problem, boundary conditions of the Dirichlet type are considered, and in the second problem, conditions of the type considered in [25] are specified.

As we noted above, direct and inverse problems for equations with involutive transformations were mainly studied for the second-order equations with one spatial variable. For high-order equations, in particular for the fourth order, as well as for equations with many space variables, such problems have not been previously studied. Similar problems have been studied only for classical equations, i.e., for equations without involutive transformations. The application of the Fourier method to the solution of such problems leads to the study of spectral questions for high-order differential equations with involutive transformations.

Previously, in [25], such questions were studied for a fourth-order equation without involution. In our research, unlike [25], we consider the fourth-order equations with involutive transformations, and we use the completeness of systems of eigenfunctions, as well as systems of eigenfunctions and associated functions for the fourth-order differential operators with involution. Studying the properties of these systems, we use the Fourier method to find solutions to the studied problems.

Let $\Omega = \{(x, y, t) : 0 < x, y < 1, 0 < t < T, T > 0\}$. Let us introduce the notations

$$\Omega_{xy} = \{(x, y) : 0 < x, y < 1\}, \Omega_{xt} = \{(x, t) : 0 < x < 1, 0 < t < T\},$$

$$\Omega_{yt} = \{(y, t) : 0 < y < 1, 0 < t < T\}.$$

Let $a_j, j = 0, 1, 2, 3$ be real numbers. We introduce the operator

$$L_4 v(x, y) \equiv a_0 \Delta^2 v(x, y) + a_1 \Delta^2 v(1 - x, y) + a_2 \Delta^2 v(x, 1 - y) + a_3 \Delta^2 v(1 - x, 1 - y).$$

where $\Delta^2 = \left( \frac{\partial^2}{\partial x^2} + \frac{\partial^2}{\partial y^2} \right)^2$ is a biharmonic operator. If $a_1 = a_3 = 0$, then instead of $L_4$, we will use the notation $L_2$. We will call the operator $L_4$ ($L_2$) a nonlocal biharmonic operator.

Note that the properties and applications of nonlocal elliptic operators $L_4$ are studied in [28–30].

Let us formulate the problems that we will use further. Consider the following problems in the domain $\Omega$.

**Problem 1.** *Find a pair of functions $\{u(x, y, t), f(x, y)\}$ such that the following conditions are satisfied:*

(1) *Functions $u(x, y, t)$ and $f(x, y)$ are smooth: $u(x, y, t) \in C(\bar{\Omega})$, $D_t^\alpha u(x, y, t), L_4 u(x, y, t) \in C(\Omega)$, and $f(x, y) \in C(\bar{\Omega})$;*

(2) *In the domain $\Omega$, they satisfy the equation*

$$D_t^\alpha u(x, y, t) + L_4 u(x, y, t) = f(x, y), (x, y, t) \in \Omega; \tag{1}$$

(3) *The following conditions are satisfied*

$$u(x, y, 0) = \varphi(x, y), u(x, y, T) = \psi(x, y), (x, y) \in \bar{\Omega}_{xy}, \tag{2}$$

$$\left. \frac{\partial^m u(x, y, t)}{\partial x^m} \right|_{x=0} = \left. \frac{\partial^m u(x, y, t)}{\partial x^m} \right|_{x=1} = 0, m = 0, 2, (y, t) \in \bar{\Omega}_{yt}, \tag{3}$$

$$\left. \frac{\partial^m u(x, y, t)}{\partial y^m} \right|_{y=0} = \left. \frac{\partial^m u(x, y, t)}{\partial y^m} \right|_{y=1} = 0, m = 0, 2, (x, t) \in \bar{\Omega}_{xt}, \tag{4}$$

*Here, $D_t^\alpha u, \alpha \in (0,1]$ is the derivative of order $\alpha$ in the sense of Caputo, i.e.,*

$$D_t^\alpha u(t,x) = \begin{cases} \frac{1}{\Gamma(1-\alpha)} \int\limits_0^t (t-\tau)^{-\alpha} u_\tau(\tau,x) d\tau, 0 < \alpha < 1 \\ \frac{du(t,x)}{dt}, \alpha = 1 \end{cases}.$$

**Problem 2.** *Find a pair of functions $\{u(x,y,t), f(x,y)\}$ such that the following conditions are satisfied:*

*(1) Functions $u(x,y,t)$ and $f(x,y)$ are smooth: $u(x,y,t) \in C(\bar{\Omega})$, $D_t^\alpha u(x,y,t), L_2 u(x,y,t) \in C(\Omega)$, and $f(x,y) \in C(\bar{\Omega})$;*

*(2) In the domain $\Omega$, they satisfy the equation*

$$D_t^\alpha u(x,y,t) + L_2 u(x,y,t) = f(x,y), (x,y,t) \in \Omega; \tag{5}$$

*(3) Conditions (1.2) are satisfied and*

$$\left.\frac{\partial^l u(x,y,t)}{\partial x^l}\right|_{x=0} = \left.\frac{\partial^l u(x,y,t)}{\partial x^l}\right|_{x=1}, \left.\frac{\partial^k u(x,y,t)}{\partial x^k}\right|_{x=0} = 0, l = 0,2, k = 1,3, (y,t) \in \bar{\Omega}_{yt}, \tag{6}$$

$$\left.\frac{\partial^k u(x,y,t)}{\partial y^k}\right|_{y=0} = \left.\frac{\partial^k u(x,y,t)}{\partial y^k}\right|_{y=1} = 0, k = 0,2, (x,t) \in \bar{\Omega}_{xt}, \tag{7}$$

*where $\varphi(x,y)$ and $\psi(x,y)$ are predefined functions.*

Note that the boundary conditions of Problem 1 are Dirichlet-type conditions, and some of the boundary conditions of Problem 2 are Samarskii–Ionkin-type conditions. In what follows, we will show that the corresponding spatial differential operator in Problem 1 is self-conjugate, whereas in Problem 2, it is non-self-conjugate. Consequently, the system of eigenfunctions corresponding to Problem 2 is incomplete. Therefore, in Problem 2, in contrast to Problem 1, it is also necessary to study the completeness and basis properties of such systems.

The results of this work are presented in the following order. In Section 2, the properties of some biorthogonal systems related to the spectral issues of Problem 2 are described. In Section 3, the known properties of the Mittag-Leffler type function, as well as a solution to a one-dimensional differential equation of a fractional order are considered. Section 4 is devoted to the study of the first inverse problem, where the main theorems on the existence and uniqueness of the solution to the studied problem are given. Section 5 presents the main theorem on the existence and uniqueness of a solution to Inverse Problem 2. The conclusion is presented in Section 6.

## 2. Study of the Properties of a Biorthogonal System

Let

$$X_0(x) = 1, X_{2n-1}(x) = \cos(2\pi nx), X_{2n}(x) = x\sin(2\pi nx), Y_k(y) = \sqrt{2}\sin(k\pi y),$$

$$\tilde{X}_0(x) = 2(1-x), \tilde{X}_{2n-1}(x) = 4(1-x)\cos(2\pi nx), \tilde{X}_{2n}(x) = 4\sin(2\pi nx), n, k \in N.$$

Consider the following systems

$$Z_{0k}(x,y) = Y_k(y), Z_{(2n-1)k}(x,y) = X_{(2n-1)}(x)Y_k(y), Z_{2nk}(x,y)$$
$$= X_{2n}(x)Y_k(y), n, k \in N, \tag{8}$$

$$W_{0k}(x,y) = \tilde{X}_0(x)Y_k(y), W_{(2n-1)k}(x,y) = \tilde{X}_{(2n-1)}(x)Y_k(y), W_{2nk}(x,y)$$
$$= \tilde{X}_{2n}(x)Y_k(y), n, k \in N. \tag{9}$$

Let us consider some well-known properties of these systems. It was shown in [25] that the functions $Z_{0k}(x,y)$, $Z_{(2n-1)k}(x,y)$ and $Z_{2nk}(x,y)$ satisfy the boundary conditions (6) and (7), while the functions $W_{0k}(x,y)$, $W_{(2n-1)k}(x,y)$, $W_{2nk}(x,y)$ satisfy conjugate conditions, i.e.,

$$\left.\frac{\partial^k W(x,y)}{\partial x^k}\right|_{x=0} = 0, k = 0, 2, \left.\frac{\partial^l W(x,y)}{\partial x^l}\right|_{x=0} = \left.\frac{\partial^l W(x,y)}{\partial x^l}\right|_{x=1}, l = 1, 3, y \in [0,1], \quad (10)$$

$$\left.\frac{\partial^k W(x,y)}{\partial y^k}\right|_{y=0} = \left.\frac{\partial^k W(x,y)}{\partial y^k}\right|_{y=1} = 0, k = 0, 2, y \in [0,1]. \quad (11)$$

In addition, the following assertions are proved (see also [27]).

**Lemma 1.** *The systems of functions (8) and (9) are biorthogonal.*

**Lemma 2.** *The systems of functions (8) and (9) form a Riesz basis in $L_2(\Omega_{xy})$.*

Systems (8) and (9) also have the following additional properties.

**Lemma 3.** *The system of functions $\left\{Z_{0k}, Z_{(2m-1)k}, Z_{2mk}\right\}$ satisfies the following equalities*

$$\Delta^2 Z_{0k}(x,y) = (k\pi)^4 Z_{0k}(x,y),$$

$$\Delta^2 Z_{(2n-1)k}(x,y) = \left[(2\pi n)^2 + (k\pi)^2\right]^2 Z_{(2n-1)k}(x,y),$$

$$\Delta^2 Z_{2nk}(x,y) = \left[(2\pi n)^2 + (k\pi)^2\right]^2 Z_{2nk}(x,y)$$
$$- 4(2\pi n)\left[(2\pi n)^2 + (k\pi)^2\right]Z_{(2n-1)k}(x,y).$$

**Proof.** For the function $Z_{0k}(x,y)$, we have

$$\Delta^2 Z_{0k}(x,y) = \left(\frac{\partial^4}{\partial x^4} + 2\frac{\partial^2}{\partial x^2}\frac{\partial^2}{\partial y^2} + \frac{\partial^4}{\partial y^4}\right)\sqrt{2}\sin(k\pi y)$$

$$= (k\pi)^4\sqrt{2}\sin(k\pi y) = (k\pi)^4 Z_{0k}(x,y).$$

Similarly, for the function $Z_{(2n-1)k}(x,y)$, we obtain

$$\Delta^2 Z_{(2n-1)k}(x,y) = X_{(2n-1)}^{(IV)}(x)Y_k(y) + 2X_{(2n-1)}''(x)Y_k''(y) + X_{(2n-1)}(x)Y_k^{(IV)}(y)$$

$$= \left[(2\pi n)^4 + 2(2\pi n)^2(k\pi)^2 + (k\pi)^4\right]X_{(2n-1)}(x)Y_k(y)$$

$$= \left[(2\pi n)^2 + (k\pi)^2\right]^2 X_{(2n-1)}(x)Y_k(y) = \left[(2\pi n)^2 + (k\pi)^2\right]^2 Z_{(2n-1)k}(x,y).$$

Additionally, at last, for $Z_{2nk}(x,y)$, we obtain

$$\Delta^2 Z_{2nk}(x,y) = X_{2n}^{(IV)}(x)Y_k(y) + 2X_{2n}''(x)Y_k''(y) + X_{2n}(x)Y_k^{(IV)}(y)$$

$$= \left[(2\pi n)^2 + (k\pi)^2\right]^2 X_{2n}(x)Y_k(y) - 4(2\pi n)\left[(2\pi n)^2 + (k\pi)^2\right]X_{2n-1}(x)Y_k(y)$$

$$= \left[(2\pi n)^2 + (k\pi)^2\right]^2 Z_{2nk}(x,y) - 4(2\pi n)\left[(2\pi n)^2 + (k\pi)^2\right]Z_{(2n-1)k}(x,y).$$

The lemma is proved. □

This lemma implies the following assertion.

**Lemma 4.** *The system of functions* $\left\{ Z_{0k}, Z_{(2m-1)k}, Z_{2mk} \right\}$ *satisfies the following equalities*

$$L_2 Z_{0k}(x,y) = \theta_{k,2}(k\pi)^4 Z_{0k}(x,y), \tag{12}$$

$$L_2 Z_{(2n-1)k}(x,y) = \theta_{k,2}\left[(2\pi n)^2 + (k\pi)^2\right]^2 Z_{(2n-1)k}(x,y), \tag{13}$$

$$L_2 Z_{2nk}(x,y) = \theta_{k,2}\left[(2\pi n)^2 + (k\pi)^2\right]^2 Z_{2nk}(x,y)$$
$$- 4(2\pi n)\theta_{k,2}\left[(2\pi n)^2 + (k\pi)^2\right] Z_{(2n-1)k}(x,y), \tag{14}$$

*where* $\theta_{k,2} = a_0 + (-1)^{k+1}a_1$.

**Proof.** As $Y_k(1-y) = (-1)^{k+1}Y_k(y)$ , from Lemma 3, it follows that

$$\Delta^2 Z_{0k}(x,1-y) = (-1)^{k+1}(k\pi)^4 Z_{0k}(x,y),$$

$$\Delta^2 Z_{(2n-1)k}(x,1-y) = (-1)^{k+1}\left[(2\pi n)^2 + (k\pi)^2\right]^2 Z_{(2n-1)k}(x,y),$$

$$\Delta^2 Z_{2nk}(x,1-y) = (-1)^{k+1}\left[(2\pi n)^2 + (k\pi)^2\right]^2 Z_{2nk}(x,y)$$

$$-4(2\pi n)(-1)^{k+1}\left[(2\pi n)^2 + (k\pi)^2\right] Z_{(2n-1)k}(x,y).$$

Hence, we obtain that equalities (12)–(14) are satisfied. The lemma is proved. □

The following assertion is proved similarly.

**Lemma 5.** *The system of functions* $\left\{ W_{0k}, W_{(2m-1)k}, W_{2mk} \right\}$ *satisfies the following equalities*

$$L_2 W_{0k}(x,y) = \theta_{k,2}(k\pi)^4 W_{0k}(x,y),$$

$$L_2 W_{(2n-1)k}(x,y) = \theta_{k,2}\left[(2\pi n)^2 + (k\pi y)^2\right]^2 W_{(2n-1)k}(x,y)$$
$$- \theta_{k,2} 4(2\pi n)\left[(2\pi n)^2 + (k\pi y)^2\right] W_{2nk}(x,y),$$

$$L_2 W_{2nk}(x,y) = \theta_{k,2}\left[(2\pi n)^2 + (k\pi)^2\right]^2 W_{2nk}(x,y).$$

## 3. On Some Properties of the Mittag-Leffler Function

Consider a one-dimensional fractional differential equation of the type

$$D_t^\alpha u(t) = \lambda u(t) + f(t), \tag{15}$$

where $0 < \alpha \le 1, \lambda \in R, f(t)$ is a given function. As is known [31], the general solution to Equation (15) is written as

$$u(t) = C \cdot E_\alpha(\lambda t^\alpha) + \int_0^t (t-\tau)^{\alpha-1} E_{\alpha,\alpha}(\lambda(t-\tau)^\alpha) f(\tau) d\tau.$$

Here and further, the symbol $C$ will denote an arbitrary positive constant whose value is not important to us; $E_{\alpha,\beta}(z)$ is a function of the Mittag-Leffler type [32], which has the form

$$E_{\alpha,\beta}(z) = \sum_{k=0}^{\infty} \frac{z^k}{\Gamma(\alpha k + \beta)}, \quad z, \alpha, \beta \in C, \ \mathrm{Re}(\alpha) > 0. \tag{16}$$

Further, we use the following properties and formulas of the Mittag-Leffler type function $E_{(}z)$:

(1) Function $E_\alpha(-z)$ for $z > 0$, $\alpha \in [0,1]$ is completely monotone [32];
(2) For $\alpha \in (0,2)$, $\gamma \le |arg z| \le \pi, \beta \in R$, $\gamma \in (\pi\alpha/2; \min\{\pi; \pi\alpha\})$ satisfies the estimate [32]

$$\left| E_{\alpha,\beta}(z) \right| \le \frac{C}{1+|z|}; \tag{17}$$

(3) The formula holds [32]:

$$E_{\alpha,\mu}(z) = \frac{1}{\Gamma(\mu)} + zE_{\alpha,\alpha+\mu}(z),$$

$$\frac{1}{\Gamma(\gamma)} \int_0^z (z-t)^{\gamma-1} E_{\alpha,\beta}(\lambda t^\alpha) t^{\beta-1} dt = z^{\beta+\gamma-1} E_{\alpha,\beta+\gamma}(\lambda z^\alpha). \tag{18}$$

## 4. Uniqueness and Existence of a Solution to Problem 1

Let $X_n(x) = \sqrt{2}\sin(n\pi x), Y_k(y) = \sqrt{2}\sin(k\pi y), n, k \in N$. Use the notation $V_{nk}(x,y) = X_n(x)Y_k(y)$. From equality (1) and conditions (3) and (4), it follows that the system $\{V_{nk}(x,y)\}_{n,k=1}^{\infty}$ are eigenfunctions of the problem

$$\Delta^2 v(x,y) = \mu^2 v(x,y), (x,y) \in \Omega_{x,y}, \tag{19}$$

$$\left.\frac{\partial^m v(x,y)}{\partial x^m}\right|_{x=0} = \left.\frac{\partial^m v(x,y)}{\partial x^m}\right|_{x=1} = 0, m = 0,2, 0 \le y \le 1, \tag{20}$$

$$\left.\frac{\partial^m v(x,y)}{\partial y^m}\right|_{y=0} = \left.\frac{\partial^m v(x,y)}{\partial y^m}\right|_{y=1} = 0, m = 0,2, 0 \le x \le 1. \tag{21}$$

The corresponding eigenvalues are $\mu_{n,k}^2 = \pi^4 [n^2 + k^2]^2, n, k \in N$.
As for the function $X_n(x), Y_k(y)$, the equalities

$$X_n(1-x) = (-1)^{n+1} X_n(x), Y_k(1-y) = (-1)^{k+1} Y_k(y),$$

are satisfied, then for $V_{nk}(x,y)$, we obtain

$$L_4 V_{nk}(x,y) = \theta_{nk,4} \mu_{nk}^2 V_{nk}(x,y), \tag{22}$$

where

$$\theta_{nk,4} = a_0 + (-1)^{n+1} a_1 + (-1)^{k+1} a_2 + (-1)^{n+k} a_3, n, k \in N. \tag{23}$$

Indeed,

$$L_4 V_{nk}(x,y) = a_0 \mu_{nk}^2 V_{nk}(x,y) + (-1)^{n+1} a_1 \mu_{nk}^2 V_{nk}(x,y) + (-1)^{k+1} a_2 \mu_{nk}^2 V_{nk}(x,y)$$

$$+ (-1)^{n+k} a_3 \mu_{nk}^2 V_{nk}(x,y) = \left[ a_0 + (-1)^{n+1} a_1 + (-1)^{k+1} a_2 + (-1)^{n+k} a_3 \right] \mu_{nk}^2 V_{nk}(x,y)$$

$$= \theta_{nk,4} \mu_{nk}^2 V_{nk}(x,y).$$

Note that for $\theta_{nk,4}$, the equalities are valid:

$$\theta_{(2n-1)(2k-1),4} = a_0 + a_1 + a_2 + a_3, \theta_{(2n-1)2k,4} = a_0 + a_1 - a_2 - a_3,$$

$$\theta_{2n2k-1,4} = a_0 - a_1 + a_2 - a_3, \theta_{2n2k,4} = a_0 - a_1 - a_2 + a_3.$$

In the case $a_2 = a_3 = 0$, we obtain $\theta_{nk,2} \equiv \theta_{n,2} = a_0 + (-1)^{n+1}a_1$, and hence,

$$\theta_{(2n-1),2} = a_0 + a_1, \theta_{2n,2} = a_0 - a_1. \tag{24}$$

Further, we will assume that $\theta_{nk,4} > 0$ for all $n, k \in N$. From equality (22) and conditions (20) and (21), it follows that $V_{nk}(x, y)$ and $\lambda_{nk} = \theta_{nk,4}\mu_{nk}^2$ are eigenfunctions and eigenvalues of the spectral problem

$$L_4 v(x, y) = \lambda v(x, y), (x, y) \in \Omega_{xy} \tag{25}$$

with boundary conditions (20) and (21).

Consider the function

$$u_{nk}(t) = \int_{\Omega_{xy}} u(x, y, t) V_{nk}(x, y) dx dy, n, k \in N. \tag{26}$$

Applying the operator $D_t^\alpha$ to equality (26), and taking into account Equation (1), we have

$$D_t^\alpha u_{nk}(t) = \langle D_t^\alpha u(x, y, t), V_{nk}(x, y) \rangle = \langle -L_4 u(x, y, t) + f(x, y), V_{nk}(x, y) \rangle$$

$$= -\theta_{nk,4}\mu_{nk}\langle u(x, y, t), V_{nk}(x, y) \rangle + \langle f(x, y), V_{nk}(x, y) \rangle = -\lambda_{nk}u_{nk}(t) + f_{nk}.$$

Moreover, from the boundary conditions (2), we obtain

$$u_{nk}(0) = \varphi_{nk}, u_{nk}(T) = \psi_{nk}, n, k \in N,$$

where

$$\varphi_{nk} = \int_{\Omega_{xy}} \varphi(x, y) V_{nk}(x, y) dx dy, \psi_{nk} = \int_{\Omega_{xy}} \psi(x, y) V_{nk}(x, y) dx dy, n, k \in N.$$

Thus, for the coefficients $u_{nk}(t)$, $n, k \in N$, we obtain the following boundary value problem

$$D_t^\alpha u_{nk}(t) = -\lambda_{nk}u_{nk}(t) + f_{nk}, \tag{27}$$

$$u_{nk}(0) = \varphi_{nk}, u_{nk}(T) = \psi_{nk}. \tag{28}$$

As we have already noted, the general solution to Equation (27) has the form

$$u_{nk}(t) = C_{nk} \cdot E_{\alpha,1}(-\lambda_{nk}t^\alpha) + \int_0^t (t - \tau)^{\alpha-1} E_{\alpha,\alpha}(-\lambda_{nk}(t - \tau)^\alpha) f_{nk} d\tau, n, k \in N,$$

where $C_{nk}, n, k \in N$ are arbitrary constants. Taking into account that $f_{nk} = Const$ and using property (18) of the function $E_{\alpha,\alpha}(-\lambda_{nk}(t - \tau)^\alpha)$ for the coefficients $u_{nk}(t)$, we obtain the representation

$$u_{nk}(t) = C_{nk}E_{\alpha,1}(-\lambda_{nk}t^\alpha) + f_{nk}t^\alpha E_{\alpha,\alpha+1}(-\lambda_{nk}t^\alpha).$$

Then, from the boundary conditions (28), we have

$$\varphi_{nk} = u_{nk}(0) = C_{nk}, \psi_{nk} = u_{nk}(T) = \varphi_{nk}E_{\alpha,1}(-\lambda_{nk}T^\alpha) + f_{nk}T^\alpha E_{\alpha,\alpha+1}(-\lambda_{nk}T^\alpha).$$

Hence, we find

$$f_{nk} = \frac{\psi_{nk} - \varphi_{nk} E_{\alpha,1}(-\lambda_{nk} T^\alpha)}{T^\alpha E_{\alpha,\alpha+1}(-\lambda_{nk} T^\alpha)}.$$

From equality (18), it follows

$$T^\alpha E_{\alpha,\alpha+1}(-\lambda_{nk} T^\alpha) = -\frac{1}{\lambda_{nk}}[E_{\alpha,1}(-\lambda_{nk} T^\alpha) - 1].$$

Then,

$$u_{nk}(t) = \varphi_{nk} E_{\alpha,1}(-\lambda_{nk} t^\alpha) + \frac{f_{nk}}{\lambda_{nk}}[1 - E_{\alpha,1}(-\lambda_{nk} t^\alpha)] \qquad (29)$$

and

$$f_{nk} = \lambda_{nk} \varphi_{nk} + \frac{\lambda_{nk}[\psi_{nk} - \varphi_{nk}]}{1 - E_{\alpha,1}(-\lambda_{nk} T^\alpha)}. \qquad (30)$$

Substituting the obtained value of into (30), we obtain the final form of the functions

$$u_{nk}(t) = \varphi_{nk} + \frac{1 - E_{\alpha,1}(-\lambda_{nk} t^\alpha)}{1 - E_{\alpha,1}(-\lambda_{nk} T^\alpha)}[\psi_{nk} - \varphi_{nk}]. \qquad (31)$$

Note that formulas (30) and (31) were obtained under the assumption that a solution to Problem 1 exists. Moreover, if conditions (2) in Problem 1 are homogeneous, i.e., $\varphi(x) \equiv 0$ and $\psi(x) \equiv 0$, then $u_{nk}(t) = 0, 0 \leq t \leq T, n, k = 1, 2, \dots$. Hence, for almost all $0 \leq t \leq T$, the condition

$$\int_{\Omega_{xy}} u(x, y, t) V_{nk}(x, y) dx dy = 0$$

is satisfied. Then, due to completeness of the system $V_{nk}(x, y)$, the equality $u(x, y, t) = 0$ holds for almost all $0 \leq t \leq T$. By the condition of the problem, $u(x, y, t) \in C(\bar{\Omega})$, and therefore, $u(x, y, t) \equiv 0, (x, y, t) \in \bar{\Omega}$. Similarly, we obtain $f(x, y) \equiv 0, (x, y) \in \bar{\Omega}_{xy}$. Hence, the solution to Problem 1 is unique. Let us formulate the main assertion for Problem 1.

**Theorem 1.** *Let the coefficients $a_j, j = 0, 1, 2, 3$ in Problem 1 be such that the conditions $\theta_{nk,4} > 0, n, k \in N$ are satisfied, and for the functions $\varphi(x, y)$ and $\psi(x, y)$, the conditions are satisfied:*

(1) $\frac{\partial^{i+j}\varphi}{\partial^i x \partial^j y}, \frac{\partial^{i+j}\psi}{\partial^i x \partial^j y} \in C(\bar{\Omega}_{xy}), i, j = \overline{0,5}, i + j \leq 6;$

(2) $\left.\frac{\partial^m \varphi}{\partial x^m}\right|_{x=0} = \left.\frac{\partial^m \varphi}{\partial x^m}\right|_{x=1} = 0, \left.\frac{\partial^m \psi}{\partial x^m}\right|_{x=0} = \left.\frac{\partial^m \psi}{\partial x^m}\right|_{x=1} = 0, m = 0, 2, 4, (y, t) \in \bar{\Omega}_{yt};$

(3) $\left.\frac{\partial^m \varphi}{\partial y^m}\right|_{y=0} = \left.\frac{\partial^m \varphi}{\partial y^m}\right|_{y=1} = 0, \left.\frac{\partial^m \psi}{\partial y^m}\right|_{y=0} = \left.\frac{\partial^m \psi}{\partial y^m}\right|_{y=1} = 0, m = 0, 2, 4, (x, t) \in \bar{\Omega}_{xt}.$

*Then, the solution to Problem 1 exists, is unique, and is represented as a series*

$$u(x, y, t) = \varphi(x, y + \sum_{n,k=1}^{\infty} \frac{1 - E_{\alpha,1}(-\lambda_{nk} t^\alpha)}{1 - E_{\alpha,1}(-\lambda_{nk} T^\alpha)}(\psi_{nk} - \varphi_{nk}) V_{nk}(x, y), \qquad (32)$$

$$f(x, y) = L_4 \varphi(x, y) + \sum_{n,k=1}^{\infty} \frac{\lambda_{nk}(\psi_{nk} - \varphi_{nk})}{1 - E_{\alpha,1}(-\lambda_{nk} T^\alpha)} V_{nk}(x, y). \qquad (33)$$

**Proof.** By its construction, the sum of the series (32) and (33) formally satisfies all the conditions of Problem 1. We only have to investigate the smoothness of the sum of these series. Let us show that $L_4 u(x, y, t) \in C(\bar{\Omega})$. Acting by the operator $L_4$ on (32) and taking into account formula (22), we have

$$L_4 u(x, y, t) = L_4 \varphi(x, y) + \sum_{n,k=1}^{\infty} \frac{1 - E_{\alpha,1}(-\lambda_{nk} t^\alpha)}{1 - E_{\alpha,1}(-\lambda_{nk} T^\alpha)} \cdot \lambda_{nk}(\psi_{nk} - \varphi_{nk}) V_{nk}(x, y). \qquad (34)$$

Let us use the notation $\Delta_{nk} = 1 - E_{\alpha,1}(-\lambda_{nk}T^\alpha)$. As $E_{\alpha,1}(0) = 1$, $\lambda_{nk} > 0$, then there is $\delta > 0$, such that $|\Delta_{nk}| \geq \delta > 0$. Then, taking into account $|V_{nk}(x,y)| \leq 2$, we obtain

$$|Lu(x,y,t)| \leq |L_4\varphi(x,y)| + C\sum_{n,k=1}^{\infty} \lambda_{nk}(|\varphi_{nk}| + |\psi_{nk}|). \tag{35}$$

Thus, the series

$$\sum_{n,k=1}^{\infty} \lambda_{nk}(|\varphi_{nk}| + |\psi_{nk}|) \tag{36}$$

is a majorant, and convergence of the series (34) reduces to the study of convergence of the series (36). Using the conditions imposed on the function $\varphi(x)$ for the coefficients $\varphi_{nk}$, we obtain

$$\varphi_{nk} = 2\int_0^1 \left(\int_0^1 \varphi(x,y)\sin n\pi x \, dx\right)\sin k\pi y \, dy = \frac{2}{n\pi}\int_0^1\int_0^1 \varphi_x(x,y)\cos n\pi x \sin k\pi y \, dx \, dy$$

$$= -\frac{2}{(n\pi)^2}\int_0^1\int_0^1 \frac{\partial^2\varphi}{\partial x^2}\sin n\pi x \sin k\pi y \, dx \, dy = -\frac{2}{(n\pi)^3}\int_0^1\int_0^1 \frac{\partial^3\varphi}{\partial x^3}\cos n\pi x \sin k\pi y \, dx \, dy$$

$$= \frac{2}{(n\pi)^4}\int_0^1\int_0^1 \frac{\partial^4\varphi}{\partial x^4}\sin n\pi x \sin k\pi y \, dx \, dy = \frac{2}{(n\pi)^5}\int_0^1\int_0^1 \frac{\partial^5\varphi}{\partial x^5}\cos n\pi x \sin k\pi y \, dx \, dy$$

$$= \frac{2}{k\pi(n\pi)^5}\int_0^1\int_0^1 \frac{\partial^6\varphi}{\partial x^5\partial y}\cos k\pi y \cos n\pi x \, dx \, dy.$$

Thus, the equality is valid:

$$\varphi_{nk} = \frac{2}{k\pi(n\pi)^5}\varphi_{nk}^{(5,1)}, \quad \varphi_{nk}^{(5,1)} = \int_0^1\int_0^1 \frac{\partial^6\varphi}{\partial x^5\partial y}\cos n\pi x \cos k\pi y \, dx \, dy. \tag{37}$$

Similarly, we obtain the equalities

$$\varphi_{nk} = \frac{2}{n\pi(k\pi)^5}\varphi_{nk}^{(1,5)}, \quad \varphi_{nk}^{(1,5)} = \int_0^1\int_0^1 \frac{\partial^6\varphi}{\partial x\partial y^5}\cos n\pi x \cos k\pi y \, dx \, dy, \tag{38}$$

$$\varphi_{nk} = \frac{2}{(k\pi)^3(n\pi)^3}\varphi_{nk}^{(3,3)}, \quad \varphi_{nk}^{(3,3)} = \int_0^1\int_0^1 \frac{\partial^6\varphi}{\partial x^3\partial y^3}\cos n\pi x \cos k\pi y \, dx \, dy. \tag{39}$$

Let us study the convergence of the series

$$\sum_{n,k=1}^{\infty} (n^4 + 2n^2k^2 + k^4)(|\varphi_{nk}| + |\psi_{nk}|). \tag{40}$$

To do this, we first examine the convergence of the series $\sum_{n,k=1}^{\infty} n^4|\varphi_{nk}|$.

Taking into account (37), applying the Cauchy–Schwarz and Bessel inequalities, we obtain

$$\sum_{n,k=1}^{\infty} n^4|\varphi_{nk}| \leq \sum_{n,k=1}^{\infty} \frac{1}{nk}|\varphi_{nk}^{(5,1)}| \leq \sqrt{\sum_{n,k=1}^{\infty} \frac{1}{n^2k^2}}\sqrt{\sum_{n,k=1}^{\infty} |\varphi_{nk}^{(5,1)}|^2} \leq C\left\|\frac{\partial^6\varphi}{\partial x^5\partial y}\right\|_{L_2(\Omega_{xy})}.$$

Hence, we conclude that the series $\sum\limits_{n,k=1}^{\infty} n^4|\varphi_{nk}|$ converges. Using conditions (38), we similarly prove the convergence of the series $\sum\limits_{n,k=1}^{\infty} k^4|\varphi_{nk}|$.

Then, from equality (39), using the Cauchy–Schwarz and Bessel inequalities, we obtain

$$\sum_{n,k=1}^{\infty} n^2 k^2|\varphi_{nk}| \leq \sum_{n,k=1}^{\infty} \frac{1}{nk}|\varphi_{nk}^{(3,3)}| \leq C\left\|\frac{\partial^6 \varphi}{\partial x^3 \partial y^3}\right\|_{L_2(\Omega_{xy})},$$

i.e., the series $\sum\limits_{n,k=1}^{\infty} n^2 k^2|\varphi_{nk}|$ converges. Hence, the series (40) also converges. Taking into account the conditions imposed on $\psi(x)$ in a similar way, we prove the convergence of the series

$$\sum_{n,k=1}^{\infty} n^4|\psi_{nk}|, \ \sum_{n,k=1}^{\infty} k^4|\psi_{nk}|, \ \sum_{n,k=1}^{\infty} n^2 k^2|\psi_{nk}|.$$

Then, according to the Weierstrass theorem, the series (34) converges absolutely and uniformly in the domain $\overline{\Omega}$, and its sum is a continuous function in this domain. Similarly, it is proved that $D_t^\alpha u(x,y,t) \in C(\bar{\Omega})$. Obviously, under the condition of the theorem, the series (32) converges and $u(x,y,t) \in C(\bar{\Omega})$. Further, we will prove that $f(x,y) \in C(\bar{\Omega}_{xy})$. From (33), taking into account that $|V_{nk}(x,y)| \leq 2$, we obtain

$$|f(x,y)| \leq |L_4\varphi(x,y)| + C\sum_{n,k=1}^{\infty} \lambda_{nk}(|\varphi_{nk}| + |\psi_{nk}|).$$

Hence, we obtain that the series (36) is also a majorant for the series (33), whose convergence, under the conditions of the theorem, was proved above. Thus, series (33) converges absolutely and uniformly in the domain $\bar{\Omega}_{xy}$, i.e., $f(x,y) \in C(\bar{\Omega}_{xy})$. The theorem is proved. □

## 5. Uniqueness and Existence of a Solution to Problem 2

In this section, we will study Problem 2. We will seek the solution to the problem in the form of series

$$u(x,y,t) = \sum_{k=0}^{\infty} u_{0k}(t)Z_{0k}(x,y) + \sum_{n,k=0}^{\infty} u_{(2n-1)k}(t)Z_{(2n-1)k}(x,y)$$

$$+ \sum_{n,k=0}^{\infty} u_{2nk}(t)Z_{2nk}(x,y), \quad (41)$$

$$f(x,y) = \sum_{k=1}^{\infty} f_{0k}Z_{0k}(x,y) + \sum_{n,k=1}^{\infty} f_{(2n-1)k}Z_{2n-1k}(x,y) + \sum_{n,k=1}^{\infty} f_{2nk}Z_{2nk}(x,y). \quad (42)$$

Here, $u_{0k}(t), u_{2n-1k}(t), u_{2nk}(t)$ are unknown functions, and $f_{0k}, f_{2n-1k}, f_{2nk}$ are unknown constants.

Using Lemma 1 for the coefficients $u_{0k}(t), u_{(2n-1)k}(t), u_{2nk}(t), f_{0k}, f_{2n-1k}, f_{2nk}$ from (41) and (42), we obtain the following representations

$$u_{0k}(t) = < u(x,y,t), W_{0k}(x,y) >, u_{(2n-1)k}(t) = < u(x,y,t), W_{(2n-1)k}(x,y) >,$$

$$u_{2nk}(t) = < u(x,y,t), W_{2nk}(x,y) >, \quad (43)$$

$$f_{0k} = < f(x,y), W_{0k}(x,y) >, f_{(2n-1)k} = < f(x,y), W_{(2n-1)k}(x,y) >,$$

$$f_{2nk} = < f(x,y), W_{2nk}(x,y) > \quad (44)$$

Then, using Equation (5) and Lemma 5 of the function $W_{0k}(x,y$ for the coefficients $u_{0k}(t)$, we obtain

$$D_t^\alpha u_{0k}(t) =< D_t^\alpha u(x,y,t), W_{0k}(x,y) >=< -L_2 u(x,y,t) + f(x), W_{0k}(x,y) >$$

$$= - < u(x,y,t), L_2 W_{0k}(x,y) > + < f(x), W_{0k}(x,y) >$$

$$= -\theta_{k,2}(k\pi)^4 < u(x,y,t), W_{0k}(x,y) > +f_{0k} = -\theta_{k,2}(k\pi)^4 u_{0k}(t) + f_{0k}.$$

In addition, from the conditions (2), it follows that

$$u_{0k}(0) =< u(x,y,0), W_{0k}(x,y) >=< \varphi(x,y), W_{0k}(x,y) >= \varphi_{0k},$$

$$u_{0k}(T) =< u(x,y,T), W_{0k}(x,y) >=< \psi(x,y), W_{0k}(x,y) >= \psi_{0k}.$$

Thus, for the function $u_{0k}(t)$, we obtain the problem

$$D_t^\alpha u_{0k}(t) + \lambda_{0k} u_{0k}(t) = f_{0k}, \tag{45}$$

$$u_{0k}(0) = \varphi_{0k}, u_{0k}(T) = \psi_{0k}, \tag{46}$$

where $\lambda_{0k} = \theta_{k,2}(k\pi)^4$.

Similarly, from Equation (5) and Lemma 5 for the coefficients $u_{2nk}(t)$, we obtain

$$D^\alpha u_{2nk}(t) + \lambda_{2nk} u_{2nk}(t) = f_{2nk}, \tag{47}$$

$$u_{2nk}(0) = \varphi_{2nk}, u_{2nk}(T) = \psi_{2nk}, \tag{48}$$

when $\lambda_{2nk} = \theta_{k,2}\left[(2\pi n)^2 + (k\pi)^2\right]^2$.

Further, using Equation (5) and Lemma 5 for the function $u_{(2n-1)k}(t)$, as in the case for the coefficients $u_{0k}(t), u_{(2n-1)k}(t)$, we obtain

$$D^\alpha u_{(2n-1)k}(t) + \lambda_{2nk} u_{(2n-1)k}(t) = f_{(2n-1)k} + \tilde{\lambda}_{2nk} u_{2nk}(t), \tag{49}$$

$$u_{(2n-1)k}(0) = \varphi_{(2n-1)k}, u_{(2n-1)k}(T) = \psi_{(2n-1)k}, \tag{50}$$

where $\tilde{\lambda}_{2nk} = 4\theta_k\left[(2\pi n)^2 + (k\pi)^2\right]$.

As follows from (15) and (16), the solution to Equation (45) that satisfies the first condition from (46) is written as

$$u_{0k}(t) = \varphi_{0k} E_\alpha(-\lambda_{0k} t^\alpha) + \int_0^t (t-\tau)^{\alpha-1} E_{\alpha,\alpha}(-\lambda_{0k}(t-\tau)^\alpha) f_{0k} d\tau.$$

Hence, taking into account $f_{0k} = Const$ and formula (18), we have

$$u_{0k}(t) = \varphi_{0k} E_\alpha(-\lambda_{0k} t^\alpha) + f_{0k} t^\alpha E_{\alpha,\alpha+1}(-\lambda_{0k} t^\alpha).$$

To find the coefficient $f_{0k}$, we use the second condition from (46). From this condition it follows that

$$\varphi_{0k} E_\alpha(-\lambda_{0k} T^\alpha) + f_{0k} T^\alpha E_{\alpha,\alpha+1}(-\lambda_{0k} T^\alpha) = \psi_{0k}.$$

Hence, we find

$$f_{0k} = \frac{1}{T^\alpha E_{\alpha,\alpha+1}(-\lambda_{0k} T^\alpha)} (\psi_{0k} - E_\alpha(-\lambda_{0k} T^\alpha)\varphi_{0k}). \tag{51}$$

Substituting the obtained values of $f_{0k}$ into the expressions for $u_{0k}(t)$, we obtain

$$u_{0k}(t) = \varphi_{0k}E_\alpha(-\lambda_{0k}t^\alpha) + \frac{t^\alpha E_{\alpha,\alpha+1}(-\lambda_{0k}t^\alpha)}{T^\alpha E_{\alpha,\alpha+1}(-\lambda_{0k}T^\alpha)}(\psi_{0k} - \varphi_{0k}E_\alpha(-\lambda_{0k}T^\alpha)). \tag{52}$$

Similarly, from (47) and (48), we find $u_{2nk}(t)$ and $f_{2nk}$. The corresponding solution has the form

$$u_{2nk}(t) = \varphi_{2nk}E_\alpha(-\lambda_{2nk}t^\alpha) + \frac{t^\alpha E_{\alpha,\alpha+1}(-\lambda_{2nk}t^\alpha)}{T^\alpha E_{\alpha,\alpha+1}(-\lambda_{2nk}T^\alpha)}(\psi_{2nk} - \varphi_{2nk}E_\alpha(-\lambda_{2nk}T^\alpha)), \tag{53}$$

$$f_{2nk} = \frac{1}{T^\alpha E_{\alpha,\alpha+1}(-\lambda_{2nk}T^\alpha)}(\psi_{2nk} - E_\alpha(-\lambda_{2nk}T^\alpha)\varphi_{2nk}). \tag{54}$$

Consider Equation (49). The solution to the equation satisfying the second boundary condition from (50) is

$$u_{(2n-1)k}(t) = \varphi_{(2n-1)k}E_\alpha(-\lambda_{2nk}t^\alpha)$$
$$+ \int_0^t (t-\tau)^{\alpha-1}E_{\alpha,\alpha}(-\lambda_{2nk}(t-\tau)^\alpha)\Big(2n\pi\tilde{\lambda}_{2nk}u_{2nk}(\tau) + f_{(2n-1)k}\Big)d\tau.$$

Taking into account that $f_{(2n-1)k} = Const$ and formula (18), we simplify the last expression

$$u_{(2n-1)k}(t) = \varphi_{(2n-1)k}E_\alpha(-\lambda_{2nk}t^\alpha) + f_{(2n-1)k}t^\alpha E_{\alpha,\alpha+1}(-\lambda_{2nk}t^\alpha)$$
$$+ 2\pi n\tilde{\lambda}_{2nk}\int_0^t (t-\tau)^{\alpha-1}E_{\alpha,\alpha}(-\lambda_{2nk}(t-\tau)^\alpha)u_{2nk}(\tau)d\tau.$$

To find the coefficient $f_{(2n-1)k}$, we use the second condition from (50). We have

$$\varphi_{(2n-1)k}E_\alpha(-\lambda_{2nk}T^\alpha) + f_{(2n-1)k}T^\alpha E_{\alpha,\alpha+1}(-\lambda_{2nk}T^\alpha)$$
$$+ 2\pi n\tilde{\lambda}_{2nk}\int_0^T (T-\tau)^{\alpha-1}E_{\alpha,\alpha}(-\lambda_{2nk}(T-\tau)^\alpha)u_{2nk}(\tau)d\tau = \psi_{(2n-1)k}.$$

Then, we obtain

$$f_{(2n-1)k} = \frac{1}{T^\alpha E_{\alpha,\alpha+1}(-\lambda_{2nk}T^\alpha)}\Big[\psi_{2n-1k} - \varphi_{2n-1k}E_\alpha(-\lambda_{2nk}T^\alpha)$$
$$- 2\pi n\tilde{\lambda}_{2nk}\int_0^T (T-\tau)^{\alpha-1}E_{\alpha,\alpha}(-\lambda_{2nk}(T-\tau)^\alpha)u_{2nk}(\tau)d\tau\Big]. \tag{55}$$

Substituting $f_{(2n-1)k}$ into the expression for $u_{(2n-1)k}(t)$, we obtain

$$u_{(2n-1)k}(t) = \varphi_{(2n-1)k}E_\alpha(-\lambda_{2nk}t^\alpha) + \frac{t^\alpha E_{\alpha,\alpha+1}(-\lambda_{2nk}t^\alpha)}{T^\alpha E_{\alpha,\alpha+1}(-\lambda_{2nk}T^\alpha)}[\psi_{2n-1k} - \varphi_{2n-1k}E_\alpha(-\lambda_{2nk}T^\alpha)]$$
$$- \frac{t^\alpha E_{\alpha,\alpha+1}(-\lambda_{2nk}t^\alpha)}{T^\alpha E_{\alpha,\alpha+1}(-\lambda_{2nk}T^\alpha)} \cdot 2\pi n\tilde{\lambda}_{2nk}\int_0^T (T-\tau)^{\alpha-1}E_{\alpha,\alpha}(-\lambda_{2nk}(T-\tau)^\alpha)u_{2nk}(\tau)d\tau$$
$$+ 2\pi n\lambda_{2nk}\int_0^t (t-\tau)^{\alpha-1}E_{\alpha,\alpha}(-\lambda_{2nk}(t-\tau)^\alpha)u_{2nk}(\tau)d\tau. \tag{56}$$

Further, using the first formula from equality (18), we represent the coefficients $u_{nk}(t)$ and $f_{nk}$ from equalities (51)–(56) as

$$u_{0k}(t) = \frac{E_\alpha(-\lambda_{0k}t^\alpha) - E_\alpha(-\lambda_{0k}T^\alpha)}{1 - E_\alpha(-\lambda_{0k}T^\alpha)}\varphi_{0k} + \frac{1 - E_\alpha(-\lambda_{0k}t^\alpha)}{1 - E_\alpha(-\lambda_{0k}T^\alpha)}\psi_{0k}, \tag{57}$$

$$f_{0k} = \frac{\lambda_{0k}}{1 - E_\alpha(-\lambda_{0k}T^\alpha)}(\psi_{0k} - E_\alpha(-\lambda_{0k}T^\alpha)\varphi_{0k}), \tag{58}$$

$$u_{2nk}(t) = \frac{E_\alpha(-\lambda_{2nk}t^\alpha) - E_\alpha(-\lambda_{2nk}T^\alpha)}{1 - E_\alpha(-\lambda_{2nk}T^\alpha)}\varphi_{2nk} + \frac{1 - E_\alpha(-\lambda_{2nk}t^\alpha)}{1 - E_\alpha(-\lambda_{2nk}T^\alpha)}\psi_{2nk}, \tag{59}$$

$$f_{2nk} = \frac{\lambda_{2nk}}{1 - E_\alpha(-\lambda_{2nk}T^\alpha)}(\psi_{2nk} - E_\alpha(-\lambda_{2nk}T^\alpha)\varphi_{2nk}), \tag{60}$$

$$\begin{aligned} u_{(2n-1)k}(t) = {} & \frac{E_\alpha(-\lambda_{2nk}t^\alpha) - E_\alpha(-\lambda_{2nk}T^\alpha)}{1 - E_\alpha(-\lambda_{2nk}T^\alpha)}\varphi_{(2n-1)k} + \frac{1 - E_\alpha(-\lambda_{2nk}t^\alpha)}{1 - E_\alpha(-\lambda_{2nk}T^\alpha)}\psi_{2n-1k} \\ & + \frac{2\pi n\tilde{\lambda}_{2nk}}{(1 - E_\alpha(-\lambda_{2nk}T^\alpha))^2}(1 - E_\alpha(-\lambda_{2nk}T^\alpha))F_{nk}(t)(\varphi_{2nk} - \psi_{2nk}) \\ & - \frac{2\pi n\tilde{\lambda}_{2nk}}{(1 - E_\alpha(-\lambda_{2nk}T^\alpha))^2}(1 - E_\alpha(-\lambda_{2nk}t^\alpha))F_{nk}(T)(\varphi_{2nk} - \psi_{2nk}), \end{aligned} \tag{61}$$

$$\begin{aligned} f_{(2n-1)k} = {} & \frac{\lambda_{2nk}}{1 - E_\alpha(-\lambda_{2nk}T^\alpha)}[\psi_{(2n-1)k} - \varphi_{2n-1k}E_\alpha(-\lambda_{2nk}T^\alpha)] \\ & - \frac{2\pi n\tilde{\lambda}_{2nk}F_{nk}(T)}{(1 - E_\alpha(-\lambda_{2nk}T^\alpha))^2}[\varphi_{2n-1k} - \psi_{2nk}] \\ & - \frac{2\pi n\tilde{\lambda}_{2nk}}{1 - E_\alpha(-\lambda_{2nk}T^\alpha)}[\psi_{2nk} - \varphi_{2n-1k}E_\alpha(-\lambda_{2nk}T^\alpha)] \end{aligned} \tag{62}$$

where

$$F_{nk}(t) = \int_0^t (t-\tau)^{\alpha-1}E_{\alpha,\alpha}(-\lambda_{2nk}(t-\tau)^\alpha)E_\alpha(-\lambda_{2nk}\tau^\alpha)d\tau.$$

Thus, the solutions to the problem have the form (41) and (42), where the functions $u_{0k}(t)$, $u_{2n-1\,k}(t)$, $u_{2n\,k}(t)$ and coefficients $f_{0k}$, $f_{2n-1k}$, $f_{2nk}$ are determined, respectively, by formulas (57)–(62) .

Let us formulate the main assertion regarding Problem 2.

**Theorem 2.** *Let coefficients $a_0$ and $a_1$ in Problem 2 be such that $a_0 \pm a_1 > 0$ and functions $\varphi(x, y)$ and $\psi(x, y)$ satisfy the conditions*

*(1)* $\quad \frac{\partial^{i+j}\varphi}{\partial^i x\partial^j y}, \frac{\partial^{i+j}\psi}{\partial^i x\partial^j y} \in C(\bar{\Omega}_{xy}), i, j = \overline{0,5}, i+j \le 6;$

*(2)* $\quad \frac{\partial^i\varphi}{\partial^i x}\Big|_{x=0} = \frac{\partial^i\varphi}{\partial^i x}\Big|_{x=1}, \frac{\partial^i\psi}{\partial^i x}\Big|_{x=0} = \frac{\partial^i\psi}{\partial^i x}\Big|_{x=1}, i = 0, 2, 4;$

*(3)* $\quad \frac{\partial^i\varphi}{\partial^i x}\Big|_{x=1} = 0, \frac{\partial^i\psi}{\partial^i x}\Big|_{x=1} = 0, i = 1, 3;$

*(4)* $\quad \frac{\partial^j\varphi}{\partial^j y}\Big|_{x=0,1} = 0, \frac{\partial^j\psi}{\partial^j y}\Big|_{x=0,1} = 0, j = 0, 2, 4.$

*Then, a solution to Problem 2 exists and is unique.*

**Proof.** The existence of a solution to the problem. Since system (8) forms the Riesz basis in the space $L_2(\Omega_{xy})$, the functions $u(x, y, t)$ and $f(x, y)$ can be represented in the form (41) and (42), where the coefficients $f_0, f_{(2n-1)k}, f_{2nk}$ and functions $u_0(t), u_{(2n-1)k}(t), u_{2nk}(t)$ are determined, respectively, by Formulas (57)–(62).

By construction, the functions $u(x, y, t)$ and $f(x, y)$ satisfy Equation (5) and conditions (6) and (7). Let us show that $L_2 u(x, y, t) \in C(\bar{\Omega})$. Taking into account Lemma 4, acting by the operator $L_2$ on (41), we have

$$
L_2 u(x, y, t) = \sum_{k=1}^{\infty} \lambda_{0k} u_{0k}(t) Z_{0k}(x, y) + \sum_{n,k=1}^{\infty} \lambda_{2nk} u_{(2n-1)k}(t) Z_{(2n-1)k}(x, y)
$$
$$
+ \sum_{n,k=0}^{\infty} u_{2nk}(t) \Big( \lambda_{2nk} Z_{2nk}(x, y) - (2\pi n) \tilde{\lambda}_{2nk} Z_{(2n-1)k}(x, y) \Big). \quad (63)
$$

Further, taking into account the following inequalities

$$
|z_{0k}(x, y)| \le \sqrt{2}, \left| z_{(2n-1)k}(x, y) \right| \le \sqrt{2}, |z_{2nk}|(x, y) \le \sqrt{2}, (x, y) \in \bar{\Omega}_{xy},
$$

we obtain

$$
|L_2 u(x, y, t)| \le C \left[ \sum_{k=1}^{\infty} \lambda_{0k} |u_{0k}(t)| + \sum_{n,k=1}^{\infty} \lambda_{2nk} \Big( |u_{(2n-1)k}(t)| + |u_{2nk}(t)| \Big) \right. \quad (64)
$$

Now, let us estimate the functions $u_{0k}(t)$, $u_{(2n-1)k}(t)$, and $u_{2nk}(t)$. From (57), we obtain

$$
|u_{0k}(t)| \le \left| \frac{E_\alpha(-\lambda_{0k} t^\alpha) - E_\alpha(-\lambda_{0k} T^\alpha)}{1 - E_\alpha(-\lambda_{0k} T^\alpha)} \varphi_{0k} \right| + \left| \frac{1 - E_\alpha(-\lambda_{0k} t^\alpha)}{1 - E_\alpha(-\lambda_{0k} T^\alpha)} \psi_{0k} \right|.
$$

Hence, taking into account the estimate (17) and the complete monotonicity of the Mittag-Leffler function, as well as the fact that $E_\alpha(0) = 1$, $\lambda_{0k} > 0$, we obtain

$$
|u_{0k}(t)| \le C(|\varphi_{0k}| + |\psi_{0k}|). \quad (65)
$$

Similarly, from (59), we obtain

$$
|u_{2nk}(t)| \le C(|\varphi_{2nk}| + |\psi_{2nk}|). \quad (66)
$$

Now, we will estimate $u_{(2n-1)k}(t)$. From (61), we obtain

$$
|u_{(2n-1)k}(t)| \le C \Big( |\varphi_{(2n-1)k}| + |\psi_{2n-1k}|
$$
$$
+ |\psi_{2n-1k}| + 2\pi n \tilde{\lambda}_{2nk}(|F_{nk}(t)| + |F_{nk}(T)|)(|\varphi_{2nk}| + |\psi_{2nk}|) \Big). \quad (67)
$$

Taking into account formulas (17) and (18), we estimate $F_{nk}(t)$

$$
|F_{nk}(t)| \le C \int_0^t (t - \tau)^{\alpha-1} E_\alpha(-\lambda_{2nk} \tau^\alpha) d\tau = C t^\alpha E_{\alpha, \alpha+1}(-\lambda_{2nk} t^\alpha) \le \frac{C}{\lambda_{2nk}};
$$

hence,

$$
|u_{(2n-1k)}(t)| \le C \left( |\varphi_{(2n-1)k}| + |\psi_{(2n-1)k}| + \frac{n}{(2\pi n)^2 + (k\pi)^2} (|\varphi_{2nk}| + |\psi_{2nk}|) \right).
$$

Then, (64) takes the form

$$|Lu(x,y,t)| \leq C\left( \sum_{k=1}^{\infty} \lambda_{0k}(|\varphi_{0k}| + |\psi_{0k}|) \right.$$

$$\left. + \sum_{n,k=1}^{\infty} \lambda_{2nk}\left(\left(\left|\varphi_{(2n-1)k}\right| + \left|\psi_{(2n-1)k}\right|\right) + (|\varphi_{2nk}| + |\psi_{2nk}|)\right)\right).$$

Hence, we obtain that the series

$$\sum_{k=1}^{\infty} k^4(|\varphi_{0k}| + |\psi_{0k}|) +$$

$$+ \sum_{n,k=1}^{\infty} \left(n^4 + n^2k^2 + k^4\right)\left(\left|\varphi_{(2n-1)k}\right| + \left|\psi_{(2n-1)k}\right| + |\varphi_{2nk}| + |\psi_{2nk}|\right) \quad (68)$$

is a majorant, and the convergence of the series (63) reduces to the study of the convergence of the series (68).

Let us show the convergence of the series

$$\sum_{k=1}^{\infty} k^4(|\varphi_{0k}| + |\psi_{0k}|). \quad (69)$$

Integrating by parts the integral in the representation of the coefficients $\varphi_{0k}$, taking into account the conditions imposed on $\varphi(x,y)$, we easily obtain the equality

$$\varphi_{0k} = \int_0^1 \int_0^1 \varphi(x,y)W_{0k}(x,y)dxdy = \frac{\sqrt{2}}{(k\pi)^5}\varphi_{0k}^5, \varphi_{0k}^5 = \int_0^1 \int_0^1 \frac{\partial^5 \varphi}{\partial y^5} \cos k\pi y dxdy. \quad (70)$$

Similarly, taking into account the conditions imposed on $\psi(x,y)$, we obtain

$$\psi_{0k} = \frac{\sqrt{2}}{(k\pi)^5}\psi_{0k}^5, \psi_{0k}^5 = \int_0^1 \int_0^1 \frac{\partial^5 \psi}{\partial y^5} \cos k\pi y dxdy. \quad (71)$$

Further, taking into account the relationship (70) and (71), and using the Cauchy–Schwarz and Bessel inequalities, we have

$$\sum_{k=1}^{\infty} k^4(|\varphi_{0k}| + |\psi_{0k}|) \leq \sum_{k=1}^{\infty} \frac{1}{k}\left(\left|\varphi_{0k}^5\right| + \left|\psi_{0k}^5\right|\right) \leq C\left(\left\|\frac{\partial^5 \varphi}{\partial y^5}\right\|_{L_2(\Omega_{xy})} + \left\|\frac{\partial^5 \psi}{\partial y^5}\right\|_{L_2(\Omega_{xy})}\right),$$

i.e., the considered series converges. Let us prove the convergence of the series

$$\sum_{n,k=1}^{\infty} n^4\left(\left|\varphi_{(2n-1)k}\right| + \left|\psi_{(2n-1)k}\right| + |\varphi_{2nk}| + |\psi_{2nk}|\right). \quad (72)$$

For coefficients $\varphi_{2n\,k}$ and $\psi_{2n\,k}$, we obtain

$$\varphi_{2n\,k} = \int_0^1 \int_0^1 \varphi(x,y)W_{2n\,k}(x,y)dxdy = \frac{4\sqrt{2}}{(2\pi n)^5 k\pi}\varphi_{2nk}^{(5,1)},$$

$$\varphi_{2nk}^{(5,1)} = \int_0^1 \int_0^1 \frac{\partial^6 \varphi}{\partial^5 x \partial y} \cos k\pi y dy \cos 2\pi nx dx, \quad (73)$$

$$\psi_{2nk} = \frac{4\sqrt{2}}{(2\pi n)^5 k\pi} \psi_{2nk}^{(5,1)}, \psi_{2nk}^{(5,1)} = \int\limits_0^1 \int\limits_0^1 \frac{\partial^6 \psi}{\partial^5 x \partial y} \cos 2\pi nx \cos k\pi y dx dy. \tag{74}$$

Similarly, for $\varphi_{(2n-1)\,k}$ and $\psi_{(2n-1)\,k}$, we obtain

$$\varphi_{(2n-1)k} = -\frac{4\sqrt{2}}{(2\pi n)^5 \pi k} \varphi_{(2n-1)k}^{(5,1)} + \frac{20\sqrt{2}}{(2\pi n)^6 \pi k} \tilde{\varphi}_{(2n-1)k'}^{(5,1)} \tag{75}$$

$$\varphi_{(2n-1)k}^{(5,1)} = \int\limits_0^1 \int\limits_0^1 \frac{\partial^6 \varphi}{\partial x^5 \partial y} (1-x) \sin 2\pi nx \cos \pi ky dx dy,$$

$$\tilde{\varphi}_{(2n-1)k}^{(5,1)} = \int\limits_0^1 \int\limits_0^1 \frac{\partial^6 \varphi}{\partial x^5 \partial y} \cos 2\pi nx \cos \pi ky dx dy, \tag{76}$$

$$\psi_{(2n-1)k} = -\frac{4\sqrt{2}}{(2\pi n)^5 \pi k} \psi_{(2n-1)k}^{(5,1)} + \frac{20\sqrt{2}}{(2\pi n)^6 \pi k} \tilde{\psi}_{(2n-1)k'}^{(5,1)} \tag{77}$$

$$\psi_{(2n-1)k}^{(5,1)} = \int\limits_0^1 \int\limits_0^1 \frac{\partial^6 \psi}{\partial x^5 \partial y} (1-x) \sin 2\pi nx \cos \pi ky dx dy,$$

$$\tilde{\psi}_{(2n-1)k}^{(5,1)} = \int\limits_0^1 \int\limits_0^1 \frac{\partial^6 \psi}{\partial x^5 \partial y} \cos 2\pi nx \cos \pi ky dx dy. \tag{78}$$

Further, taking into account the relation (73)–(79) and using the Cauchy–Schwarz and Bessel inequality, we obtain

$$\sum_{n,k=1}^{\infty} n^4 \left( \left| \varphi_{(2n-1)k} \right| + \left| \psi_{(2n-1)k} \right| + |\varphi_{2nk}| + |\psi_{2nk}| \right)$$

$$\leq \sum_{n,k=1}^{\infty} \frac{1}{nk} \left( \left| \varphi_{(2n-1)k}^{(5,1)} \right| + \left| \psi_{(2n-1)k}^{(5,1)} \right| + \left| \varphi_{2nk}^{(5,1)} \right| + \left| \psi_{2nk}^{(5,1)} \right| \right)$$

$$\leq C \left( \left\| \frac{\partial^6 \varphi}{\partial x^5 \partial y} \right\|_{L_2(\Omega_{xy})} + \left\| \frac{\partial^6 \psi}{\partial x^5 \partial y} \right\|_{L_2(\Omega_{xy})} \right),$$

i.e., the considered series converges.

The convergence of the remaining series is proved similarly. Hence, the series (68) majorizing the functional series (63) converges. Then, according to the Weierstrass theorem, the series (63) converges absolutely and uniformly in the domain $\overline{\Omega}$, and its sum is a continuous function in this domain. It is proved similarly that $_C D_{0t}^\alpha u(x,y,t) \in C(\bar{\Omega})$, and the condition $f(x,y) \in C(\overline{\Omega}_{xy})$ follows from Equation (5) and from the fact that $_C D_{0t}^\alpha u(x,y,t), L_2 u(x,y,t) \in C(\overline{\Omega})$.

Uniqueness of the solution. Suppose the opposite. Let Problem 2 have two different solutions $\{u_1(x,y,t), g_1(x,y)\}, \{u_2(x,y,t), g_2(x,y)\}$ and

$$u(x,y,t) = u_1(x,y,t) - u_2(x,y,t), f(x,y) = f_1(x,y) - f_2(x,y).$$

Then, it is easy to check that $(u(x,y,t), f(x,y))$ satisfies the Equation (5), conditions (6) and (7), and

$$u(x,y,0) = 0, u(x,y,T) = 0, (x,y) \in \bar{\Omega}_{xy}. \tag{79}$$

We will show that problems (5)–(7) and (79) have only a trivial solution. Let $(u(x, y, t), f(x, y))$ be a solution of this problem. Taking into account that $\varphi(x, y) = 0$, $\psi(x, y) = 0$, and Lemma 2, from the equalities (45), (46), (51), and (54), we obtain

$$\varphi_{0k}(t) = \psi_{0k} = 0, \ \varphi_{(2n-1)k}(t) = \psi_{(2n-1)k} = 0, \varphi_{2nk}(t) = \psi_{2nk} = 0.$$

Then, from (61)–(66), it follows that

$$u_{0k}(t) = u_{(2n-1)k}(t) = u_{2nk}(t) = 0,$$

$$f_{0k} = f_{(2n-1)k} = f_{2nk} = 0.$$

Using these values in equalities (43) and (44) , we obtain that the functions $u(x, y, t)$ and $f(x, y)$ are orthogonal to system (9), which is complete and forms a basis in $L_2(\Omega_{xy})$. Hence, almost everywhere, the equalities $u(x, y, t) = 0$ in $\Omega$ and $f(x, y) = 0$ in $\Omega_{xy}$ are correct. Since $u(x, y, t) \in C(\bar{\Omega})$ and $f(x, y) \in C(\bar{\Omega}_{xy})$, we conclude that $u(x, y, t) \equiv 0$ and $f(x, y) \equiv 0$, respectively, in the domains $\bar{\Omega}$ and $\bar{\Omega}_{xy}$, i.e., $u_1(x, y, t) \equiv u_2(x, y, t)$ and $f_1(x, y) \equiv f_2(x, y)$. The theorem is proved.  □

## 6. Conclusions

In this paper, for some classes of fourth-order differential equations with involution, we studied the solvability of inverse problems aimed at determining the right-hand side, depending on the spatial variable. Two types of problems are considered: the first problem with boundary conditions of the Dirichlet type and the second problem with conditions of the Samarskii–Ionkin type. To study these problems, we applied the method of separation of variables. When studying the first problem, we used the properties of eigenfunctions of similar problems for the second-order operators. In the study of the second problem, the properties of eigenfunctions and associated functions of spectral problems for the second-order operators were used. Using the properties of these systems, we proved the theorems on the existence and uniqueness of solutions to the studied problems.

The same method can be used to study similar problems for equations of a higher order, as well as for equations with multiple involutions. Our further investigations will be directed to the study of such problems.

**Author Contributions:** M.M., B.K., M.K. and B.T. contributed equally to the conceptualization and the formal analysis of the problem discussed in the paper. All authors have read and agreed to the published version of the manuscript.

**Funding:** The work was supported by a grant from the Ministry of Science and Education of the Republic of Kazakhstan (grant no. AP09259074).

**Institutional Review Board Statement:** Not applicable.

**Informed Consent Statement:** Not applicable.

**Data Availability Statement:** All the data is present within the manuscript.

**Conflicts of Interest:** The authors declare no conflict of interest.

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
