# Peer review of "On Solvability of Some Inverse Problems for a Fractional Parabolic Equation with a Nonlocal Biharmonic Operator"

_fractalfract, doi:10.3390/fractalfract7050404_

Round 1

Reviewer 1 Report

Report on “On solvability of some inverse problems for a fractional
parabolic equation with a nonlocal biharmonic operator” by M. Muratbekova, B. Kadirkulov, M. Koshanova and B. Turmetov, submitted for possible publication in Fractal and Fractional.

In the paper under review, the authors investigated the solvability result for a class of inverse problems involved with fractional differential equations with nonlocal biharmonic operators. For the main results, they applied standard techniques including Fourier method to solve the considered system.

The paper is well written and organized, the proofs look correct and the results are sufficiently original. However, few changes should be made prior publication.

1/ In Abstract, check “The properties of eigenfunctions, as well as eigenfunctions”.

2/ write at the Introduction section clearly the motivation and novelties of this paper with respect to the existing literature.

3/ Cite Lemmas 1,2,3,5 to indicate their useful.

4/ Page 5, above line 107, in the general solution to Eq. (21). What is “C”? is it initial condition for Eq. (21)?

5/ indicate the difference between Problem 1 and Problem 2.

6/ For the Proof of Theorem 2, it is needed to start with the existence result then the uniqueness. Also, the proof of uniqueness result is not clear since we usually consider two solutions then comparing between them.

Author Response

For Reviewer 1.

Thank you very much for your comments and suggestions. All comments and suggestions are taken into account. Below are the comments and answers to them, as well as corrections. Changes made to the manuscript are highlighted in a different color. 

Responses to the comments of  Reviewer 1.

Comment 1.

1/ In Abstract, check “The properties of eigenfunctions, as well as eigenfunctions”.

Answer: We agree with the reviewer and made appropriate changes.

 In the Abstract the sentence “ The properties of eigenfunctions, as well as eigenfunctions and associated functions of the corresponding spectral problems are studied ”

was replaced by

“The properties of eigenfunctions and associated functions of the corresponding spectral problems are studied.”

Comment 2.

Write at the Introduction section clearly the motivation and novelties of this paper with respect to the existing literature.

Answer: We agree with the reviewer and added the following text

As we noted above, direct and inverse problems for equations with involutive transformations were mainly studied for the second-order equations with one spatial variable. For high-order equations, in particular for the fourth order, as well as for equations with many space variables, such problems have not been previously studied. Similar problems have been studied only for classical equations, i.e. for equations without involutive transformations.

The application of the Fourier method to the solution of such problems leads to the study of spectral questions for high-order differential equations with involutive transformations.

Previously, in [21], such questions were studied for a fourth-order equation without involution. In our research, unlike [21], we consuder the fourth-order equations with involutive transformations.

Comment 3.

Cite Lemmas 1,2,3,5 to indicate their useful.

We agree with the reviewer and made appropriate changes.

In the main part of the article, we have quoted all the Lemmas indicated by the reviewer.

Comment 4.

Page 5, above line 107, in the general solution to Eq. (21). What is “C”? is it initial condition for Eq. (21)?

Answer: We agree with the reviewer and made appropriate changes.

We added the following sentence.

Here and further, the symbol C will denote an arbitrary positive constant whose value is not important to us.

Comment 5.

Indicate the difference between Problem 1 and Problem 2.

Answer: We agree with the reviewer, and after setting Problem 2, we added a paragraph, where we showed the difference between Problem 1 and Problem 2.

We added the following text:

Note that the boundary conditions of Problem 1 are Dirichlet-type conditions, and some of the boundary conditions of Problem 2 are Samarskii-Ionkin-type conditions. In what follows, we will show that the corresponding spatial differential operator in Problem 1 is self-conjugate, whereas in Problem 2 it is non-self-conjugate. Consequently, the system of eigenfunctions corresponding to Problem 2 is incomplete. Therefore, in Problem 2, in contrast to Problem 1, it is also necessary to study the completeness and basis properties of such systems.

Комментарий 6.

 Для доказательства теоремы 2 необходимо начать с результата существования, а затем с единственности. Кроме того, результат доказательства уникальности не ясен, так как мы обычно рассматриваем два решения, а затем сравниваем их.

Ответ: Мы соглашаемся с рецензентом и вносим соответствующие изменения.

Мы изменили пункты доказательств теоремы 2.

Reviewer 2 Report

The paper requires a thorough review of its English, as well as mathematical structure. The tautologies like "If a_2=a_3=0, then we will use the notation
a_2=a_3=0" (on Page 2) must be eliminated. "sin" in the beginning of Section 2 must be in the same format. What does it mean "quite monotonous" on the bottom of Page 5? Does it mean "completely monotone"? Please, change the word "Investigation" in titles of Sections 4 and 5. These types of remarks can be continued, bu I stop here, assuming authors will thoroughly revise their manuscript.

Author Response

For Reviewer 2.

Thank you very much for your comments and suggestions. All comments and suggestions are taken into account.  Below are the comments and answers to them, as well as corrections. Changes made to the manuscript are highlighted in a different color. 

Responses to comments of reviewer 2.

Comment 1.

The tautologies like "If a_2=a_3=0, then we will use the notation a_2=a_3=0" (on Page 2) must be eliminated.

Answer: We agree with the reviewer and made appropriate changes.

We added the following text.

If , then instead of we will use the notation . We will call the operator  ()  a nonlocal biharmonic operator.

Comment 2.

"sin" in the beginning of Section 2 must be in the same format.

Answer: We agree with the reviewer and have corrected the typo.

          We have changed the format of the formula.

Comment 3.

What does it mean "quite monotonous" on the bottom of Page 5? Does it mean "completely monotone"?

Answer: We agree with the reviewer and made appropriate changes.

In line 110 “quite monotonous” was replaced by “completely monotone”

Comment 4.

Please, change the word "Investigation" in titles of Sections 4 and 5.

Answer: We agree with the reviewer and made appropriate changes.

We have changed the subtitles.

Uniqueness and Existence of a Solution to Problem 1.

Uniqueness and Existence of a Solution to Problem 2.

Comment 5.

These types of remarks can be continued, bu I stop here, assuming authors will thoroughly revise their manuscript.

Answer: We agree with the reviewer.We revised our manuscript and made appropriate changes.
